# Clinical Significance and Remaining Issues of Anti-HBc Antibody and HBV Core-Related Antigen

**DOI:** 10.3390/diagnostics14070728

**Published:** 2024-03-29

**Authors:** Yoshihiko Yano, Itsuko Sato, Takamitsu Imanishi, Ryutaro Yoshida, Takanori Matsuura, Yoshihide Ueda, Yuzo Kodama

**Affiliations:** 1Division of Gastroenterology, Department of Internal Medicine, Kobe University Graduate School of Medicine, Kobe 650-0017, Japan; yoshidar@med.kobe-u.ac.jp (R.Y.); tmatsu@med.kobe-u.ac.jp (T.M.); yueda@med.kobe-u.ac.jp (Y.U.); kodama@med.kobe-u.ac.jp (Y.K.); 2Department of Clinical Laboratory, Kobe University Hospital, Kobe 650-0017, Japan; itsuko@med.kobe-u.ac.jp (I.S.); imanisi@med.kobe-u.ac.jp (T.I.)

**Keywords:** anti-HBc antibody, HBV core-related antigen, HBV

## Abstract

Currently, hepatitis B virus (HBV) core antibody (anti-HBc antibody) and HBV core-related antigen (HBcrAg) are widely used as serum markers for diagnosis based on the HBV core region. This review focused on anti-HBc antibodies and HBcrAg and aimed to summarize the clinical significance of currently used assay systems and the issues involved. While anti-HBc is very significant for clinical diagnosis, the clinical significance of quantitative assay of anti-HBc antibody has been reevaluated with improvements in diagnostic performance, including its association with clinical stage and prediction of carcinogenesis and reactivation. In addition, concerning the new HBcrAg, a high-sensitivity assay method has recently been established, and its diagnostic significance, including the prediction of reactivation, is being reevaluated. On the other hand, the quantitative level of anti-HBc antibody expressed in different units among assay systems complicates the interpretation of the results. However, it is difficult to standardize assay systems as they vary in advantages, and caution is needed in interpreting the assay results. In conclusion, with the development of highly sensitive HBcrAg and anti-HBc antibody, a rapid and sensitive detection assay system has been developed and used in clinical practice. In the future, it is hoped that a global standard will be created based on the many clinical findings.

## 1. Introduction

The genome of hepatitis B virus (HBV) is an incomplete double-stranded circular DNA in which the minus strand (long strand) has a total length of approximately 3.2 kb and 15–50% of the 3′-end of the plus strand (short strand) is lost. Four open reading frames (ORFs), i.e., the Pre-S/S gene, Pre-C/C gene, P (Pol) gene, and X gene, are present on the minus strand with partial overlapping [1]. Of the Pre-C/C gene, the C gene produces a core protein (HBc antigen) consisting of 183 to 185 amino acids. A total of 120 dimers of the core protein assemble to form a capsid. The C-terminus of the core protein contains an arginine-rich sequence, and the core protein binds to the viral nucleic acid (pregenomic RNA) at this site to form a nucleocapsid [2,3]. The Pre-C region encoding 29 amino acids is connected upstream of the C gene in the same frame, and a pre-core precursor protein is produced from the Pre-C/C gene. Although the pre-core region is not essential for viral replication, it is thought to play an important role in evading host immunity (especially immunity against the core protein). HBe antigen detected in serum has clinical significance as an indicator of HBV infectivity, and the epitope of HBc antigen is considered an important region involved in cellular and humoral immune responses such as T cell proliferation and the production of anti-HBc antibody [4].

The diagnostic significance of blood tests is changing with the development of many HBV assay systems and changes in disease management. This review focused on anti-HBc antibodies and HBcr antigens and aimed to summarize the clinical significance of currently used assay systems and the issues involved.

## 2. Assay and Clinical Usage of Anti-HBc Antibody

### 2.1. Clinical Significance of Anti-HBc Antibody

Since anti-HBc antibodies appear relatively early in HBV infection and is maintained almost throughout life, it is a serological indicator widely used for detecting HBV-infected persons, including those infected previously. For this reason, it has traditionally been considered useful for checking the safety of blood transfusion [5].

In recent years, quantitative anti-HBc antibody testing has become widespread, and many clinical findings have been reported regarding the assay results. Among anti-HBc antibodies, there is the anti-IgM type, which increases particularly in acute hepatitis B, and the anti-IgG type, which increases in chronic persistent infection. The significance of the anti-IgG-HBc antibody in acute hepatitis B (AHB) is not very clear, since its elevation is followed by the appearance of the anti-IgG-HBc antibody. Interestingly, using a mouse model, Wang et al. reported that the anti-IgG-HBc antibody, which appears in an early stage of HBV infection, suppresses the increase in HBs antigen [6]. While the anti-IgM type HBc antibody also increases in the acute exacerbation period of chronic hepatitis, the titer is not considered high. It is reported that IgM anti-HBc was significantly higher in AHB, while HBV DNA level was significantly higher in acute exacerbation of chronic hepatitis B (CHB-AE), and IgM-HBc level of ≥8 S/CO (sample to the cut-off value) and an HBVDNA level of <5.5 logIU/mL are considered useful for the differentiation of AHB and CHB-AE (sensitivity 98.1%, specificity 86.2%) [7]. Lall et al. also reported from India that an anti-IgM-HBc antibody level with an S/CO of ≥20.5 indicated AHB with a sensitivity of 93.3% and a specificity of 92.7% [8].

### 2.2. Assay and Units Using for Quantitative Anti-HBc Antibody

The international reference standard is expressed in terms of IU/mL, but the current anti-HBc antibody assay has shifted from the competitive method to the double-antigen sandwich assay with improvements in sensitivity. The anti-HBc antibody is assayed by techniques including enzyme-linked immunosorbent assay (ELISA), chemiluminescent microparticle immunoassay (CMIA), chemiluminescent immunoassay (CLIA), and chemiluminescent enzyme immunoassay (CLEIA) depending on the instrument used, and the results are presented in units such as IU/mL, INH%, and S/O, etc., so caution is needed in interpreting the assay results [9,10,11,12,13,14,15,16,17,18,19,20] (Table 1). The ORTHO^®^ HBc ELISA test system (Ortho-Clinical Diagnostics, Raritan, NJ, USA) has been used for a long time and shows good agreement in sensitivity and specificity with other assays [13]. MONOLISA anti-HBc EIA (Bio-Rad, Hercules, CA, USA) is also used for epidemiological data in developing countries such as those in Africa, because it is an EIA method relatively easy to handle [12]. ADVIA Centaur^®^ (Bayer Diagnostics, Tarrytown, NY, USA) has been reported by Helden and Dati et al. to be useful since 2004 [9,10], and Cavalieri et al. also demonstrated high sensitivity and concordance rates using Murex and MONOLISA [11]. Some conventional anti-HBc antibody assay systems detect both IgM and IgG types, requiring caution. To clarify the difference between IgG and IgM types, reagents that detect only the IgG type have recently been developed. With Lumipulse HBcAb-II^®^ (Fujirebio, Tokyo, Japan), which detects both the IgM and IgG types, the differentiation of acute and chronic hepatitis used to be considered difficult, but newer Lumipulse HBcAb-N^®^ is reported to detect IgG antibody alone and to be unaffected by the IgM antibody [14]. In January 2014, ADVIA Centaur^®^ HBc Total 2 (HBcT2) was approved by the FDA and has been shown to be effective for differentiation between IgG and IgM types (available at https://www.accessdata.fda.gov/cdrh_docs/pdf21/P210019B.pdf (accessed on 20 March 2024)).

All reagents widely used in the world today have sufficiently high sensitivity and specificity. Schmid et al. compared Elecsys^®^ (Roche Diagnostics, Basel, Switzerland) and ARCHITECT^®^ (Abbott Laboratories, Chicago, IL, USA) assays using donated blood samples and found them useful for screening with a sensitivity of 100% and specificity of more than 95.5% [16]. Hottenträger et al. examined the sensitivity and specificity of the Elecsys^®^ and ARCHITECT^®^ assays in German samples and reported that both systems were useful, showing 99% or higher sensitivity and specificity [21]. Maugard et al. also studied the usefulness of Elecsys^®^ in French donated blood samples and reported a good detection rate [22]. Hourfar and Schmidt et al. compared available HBc antibody assay systems and reported that the sensitivity and specificity of each kit were acceptable [16]. Li et al. used Wantai’s Diagnostics Kit (Wantai Biological Pharmacy Enterprise, Beijing, China) for antibody to hepatitis core antigen (ELISA) to confirm the presence of HBc antibody in various regions of China and assessed the risk of infection associated with blood transfusion [23].

Highly sensitive assay systems are being developed, but there is a risk of clinical meaninglessness or unnecessary anxiety for the examiner if the test result is a false positive. A German study of donated blood specimens using three types of HBc antibody tests (CMIA, EFLA, and ELISA) identified 117 of 109,603 specimens as “false positive” or “specificity confirmed” [24]. It is generally difficult to distinguish between false positives and true negatives in samples with low-titer positives using quantitative testing, and improving the accuracy of assays is a future challenge. According to a report from Iran, it was reported that the response of anti-HBs antibody after HBV vaccination in previously infected individuals with isolated anti-HBc antibody positive was lower those with true-negative individuals [25]. However, this finding is difficult to interpret because of the difference in age backgrounds.

### 2.3. Isolated Anti-HBc Antibody Positive

Among anti-HBc antibody-positive carriers, the HBsAg-negative and anti-HBs antibody-negative population is known as isolated anti-HBc (IAHBc) [24]. IAHBc can be a risk for HBV reactivation after organ transplantation, chemotherapy, or immunosuppressive therapy [25,26,27]. IAHBc is sometimes a risk for viral reactivation in immunosuppressive patients and is relatively common in dialysis and human immunodeficiency virus (HIV)-infected patients [28,29,30]. Regarding the risk of IAHBc, there has been a report that viral mutations, such as sI92T, sQ129H, rtL80I, rtS85F, and rtL91I, are related [31], and a case–control study involving Indonesians suggested an involvement of genetic polymorphism of HLA-DP of the host [32]. The clinical importance of IAHBc has been suggested by a recent cohort study involving 609,299 South Koreans, which reported that IAHBc was observed in 3.8% of the cohort and was a risk factor for liver-related death [33].

### 2.4. Anti-HBc Antibody in Patients with Chronic Hepatitis

Concerning the treatment for chronic hepatitis, there are a number of reports that the anti-HBc antibody level is related to the treatment response. Liao et al. reported that the risk of hepatitis increased when the anti-HBc antibody level was above a cut-off level of ≥4.0 log IU/mL in untreated CHB patients [18]. Fang et al. reported that the anti-HBc antibody level is useful for the prediction of response to interferon treatment in HBe antigen-positive CHB patients [34]. Wang et al. investigated the clearance of HBs antigen in patients treated with a nucleic acid analogue preparation added to PEG-IFN and reported that the clearance rate of HBs antigen was high in patients with a low pretreatment HBc antibody level [35]. Zhao et al. examined whether there was seroconversion of HBe antigen after the beginning of treatment with a nucleic acid analogue and found that patients who showed serological response (HBe seroconversion) had a significantly higher pretreatment HBc antibody level [36].

Recently, possible criteria for the discontinuation of nucleic acid analogue therapy have been discussed, and high anti-HBc antibody (anti-HBc ≥ 1000 IU/mL: HR 0.31 per log IU/m) and low HBs antigen (HBsAg < 100 IU/mL: HR 1.71 per log IU/m) are suggested to be factors of reduced clinical relapse [37].

Also, the changes in anti-HBc antibody titer during CHB-AE have not been sufficiently investigated. A recent report from China suggests that anti-HBc antibody is also useful for the prediction of prognosis of ACLF (acute-on-chronic liver failure). In this report, patients with a low anti-HBc antibody level at the onset of ACLF had a significantly worse prognosis [38]. There is room for further study on changes in the anti-HBc antibody level and the clinical significance in AHB and CHB-AE.

According to a report from Hong Kong, a cut-off value of 82.50 COI for anti-HBc antibody using Limipulse is a useful predictor of negative conversion of HBs antigen within 1 year [39]. A report from Taiwan also indicated that an anti-HBc antibody level of <3 log IU/mL is related to long-term seroclearance of HBs antigen [40].

### 2.5. Anti-HBc Antibody Positivity as a Risk of Carcinogenesis

The relationship between anti-HBc antibody positivity and the risk of HCC is controversial. In 2010, Ohki et al. reported that the effect of anti-HBc antibody positivity was not statistically significant in HCV patients after exclusion of the effects of confounding factors such as male sex and age [41]. In a Japanese study, Tsubouchi et al. found no effect of anti-HBc antibody on HCC or liver-related deaths in HCV-positive individuals either [42]. In 2011, it was reported from the USA that anti-HBc antibody positivity had no effect on carcinogenesis in hepatitis C patients [43]. On the other hand, in 2016, a meta-analysis of 26 original studies reported an increased risk of HCC with an HR of 1.36 for HBs antibody-positive/HBc antibody-positive cases and an HR of 2.15 for HBs antibody-negative/HBc antibody-positive cases compared to HBs antibody-negative/HBc antibody-negative individuals [44]. Among 8513 HCV patients from a UK database, anti-HBc antibody positivity was also reported to increase the risk of liver cirrhosis with an HR of 1.29 and of HCC with an HR of 1.64 [45].

In a report from Hong Kong, Chan et al. reviewed 489 cases of NAFLD and 69 cases of NAFLD-related cryptogenic HCC. According to the report, in NAFLD cases, fibrosis was significantly more advanced when they were anti-HBc antibody-positive (cirrhosis was observed in 18.8% of anti-HBc antibody-positive cases and 7.5% of anti-HBc antibody-negative cases), and a high percentage (73.9%) of HCC cases were anti-HBc antibody-positive [46]. From Egypt, a country where HCV is endemic, El-Maksoud et al. reported that genotype D was the major genotype and that anti-HBc antibody-positive cases were significantly younger and showed a higher histological grade of fibrosis [47]. Although there are various reports, race and ethnicity play major roles in the development of HCC, and further collection of information is considered necessary [48].

### 2.6. Transplantation and Anti-HBc Antibody

In 1998, Uemoto et al. reported that positive conversion of HBs antigen was observed in 16 of the 17 anti-HBs antibody-negative recipients of liver transplantation from anti-HBc antibody-positive donors [49]. Concerning liver transplantation, Joya-Vazquez et al. reported in 2002 that the risk of HBV reinfection was 2.5-fold higher with the use of anti-HBc antibody-positive grafts and that the risk of reinfection was reduced to one quarter with treatment by lamivudine and HBIG [50]. According to a systematic review in 2010, transplantation from anti-HBc antibody-positive donors is basically safe, but HBV reinfection was observed in 11% of the recipients, and HBIG and lamivudine are recommended for anti-HBc antibody-negative recipients [51]. The authors further reported that de novo hepatitis occurred in 19% of liver transplantations from anti-HBc antibody-positive donors to HBs antigen-negative recipients, and that reactivation was observed in 15% of anti-HBc/HBs antibody-positive recipients and in 48% of HBV naïve recipients without prophylaxis, but that the risk of reactivation was greatly reduced under prophylaxis with HBIG, lamivudine, or their combination. On the other hand, the use of lamivudine and HBIG was shown to be associated with a high frequency of escaping mutations in anti-HBc antibody-negative recipients, and next-generation nucleic acid analogues are recommended [52]. Ueda et al. also reported that escape mutations were observed in 7 of 19 cases of de novo hepatitis and that lamivudine alone was not sufficient [53]. Subsequent advances in antiviral agents have largely eliminated the problem of antiviral-induced mutations after liver transplantation [54]. In 2019, a report from Hong Kong also evaluated long-term post-transplant outcomes of liver transplantations from 461 anti-HBc antibody-positive and 548 anti-HBc antibody-negative donors and showed that there was no difference in long-term prognosis, that de novo hepatitis observed in 3 of the 108 anti-HBc antibody-negative recipients all occurred during the time when lamivudine was used, and that such instances have recently been nearly eliminated [55]. In a report from South Korea in 2021, the long-term prognoses of liver transplantation from 457 anti-HBc antibody-positive and 898 anti-HBc antibody-negative donors were examined, and even anti-HBc antibody-positive grafts were shown to have no impact on long-term survival. The same paper also reported that de novo hepatitis was observed in 0.9% of the transplants from anti-HBc antibody-positive donors to anti-HBc antibody-negative recipients [56].

Due to the appropriate immunosuppressive regimen including nucleot(s)ide analogue and immunoglobulin, kidney transplantation is now possible even for HBs antigen-positive donors after risk assessment [57]. For kidneys of HBs antigen-negative and HBc antibody-positive donors, the risk of reactivation is not high and the need for antiviral therapy has not been demonstrated [58,59]. There was a study that evaluated HBV reactivation after renal transplantation in 631 HBs antigen-negative HBc antibody-positive recipients. Reactivation was observed in two cases, and although its frequency is low, caution is warranted [28]. On the other hand, in hematopoietic stem cell transplantation, prophylaxis therapy should be introduced if the recipient is anti-HBc antibody-positive. In addition, regular follow-up is required even if the patient is HBs antigen- and HBc antibody-negative [60].

### 2.7. Reactivation and Seroconversion during Immunosuppressive State

When HBs antigen is eliminated in HBV carriers, anti-HBc antibody is the only serum marker for the detection of previous infection. Since fulminant hepatitis from occult HBV infection (OBI) is sometimes fatal [61,62], measurement of anti-HBc and anti-HBs antibodies along with HBs antigen is mandatory before immunosuppressive therapy [63,64,65]. In the treatment of malignant lymphoma also, the higher the value on pretreatment HBc antibody testing, the higher the risk of reactivation in HBs antigen-negative anti-HBc antibody-positive patients [66]. Although immune checkpoint inhibitors (ICIs) have been used for the treatment of many cancers in recent years, as of now, no OBI reactivation due to ICIs has been reported [67,68]. Clarke et al. used anti-TNF preparations in 120 HBs antigen-negative anti-HBc antibody-positive cases and found that positive conversion of HBs antigen occurred in only one case (0.8%), indicating a low frequency of reactivation [69].

In general, a person becomes anti-HBc antibody-positive throughout life after HBV infection, but both anti-HBc and anti-HBs antibodies may convert to negative under immunosuppressive conditions. Holtkamp et al. reported that permanent or intermittent anti-HBc loss was observed in 139 of 120,531 subjects by examination using Architect and that most of them were hematologic or solid tumor patients, organ transplant recipients, or HIV patients [70].

For anti-HBc antibody-positive patients, appropriate monitoring is required to prevent reactivation during anticancer and immunosuppressive therapies. The use of new agents, such as biologics and immune checkpoint inhibitors, has made it more important than ever to properly identify previously infected patients. Further data are also needed on negative conversion of anti-HBc antibody in the natural course of HBV infection.

## 3. Assay and Clinical Usage of HBV Core-Related Antigen

### 3.1. Basics of HBV Core-Related Antigen

The HBc antigen cannot be detected in blood because it is wrapped in the outer shell (envelope) and resides inside the HBV particle. HBcrAg is HBc antigen made more sensitively detectable together with the HBe antigen inside the core particle by degrading HBV core particles in blood to the smallest protein units (peptides). HBcrAg is the generic term for three antigenic component proteins: HBc antigen translated from pregenomic mRNA (hepatitis B core antigen), HBe antigen translated from precore mRNA (hepatitis B e antigen), and p22cr antigen (the 22 kDa procure protein) [71,72]. The HBcrAg assay system detects these three antigens together [73], and the results obtained have been shown to correlate well with hepatic cccDNA (covalently circular closed DNA), HBVDNA, and HBs antigen levels [74,75,76,77,78].

A European research group mainly based in France also examined tissue samples from untreated chronic hepatitis B patients by Lumipulse G HBcrAg assay using the LUMIPULSE G1200 Analyzer (Fujirebio Europe, Gent, Belgium) and reported that serum HBcrAg correlates with hepatic HBVDNA, pgRNA, cccDNA, and transcriptional activity, indicating that HBcrAg can be an indicator of both hepatic cccDNA and transcriptional activity [79].

Recently, high-sensitivity assaying of HBcrAg (iTACT-HBcrAg, cut-off value: 2.1 logIU/mL) has become possible [80]. By using a high-sensitivity system, the assay limit, which was previously 3.0–7.0 logU/mL, has been improved to 2.1–3.0 log U/mL (Table 2).

### 3.2. Clinical Staging of Chronic Hepatitis and HBcrAg

The significance of HBcrAg in acute hepatitis has not been fully elucidated, but it has recently been reported that the kinetics of HBcrAg during acute hepatitis correlate roughly with HBe antigen and HBVDNA [81].

In the natural course of chronic hepatitis B, the disease progresses through the phases of immune tolerance (IT), immune clearance (IC), HBeAg-negative inactive/quiescent carrier (ENQ), and HBeAg-negative hepatitis (ENH). HBcr antigen has been shown to decrease with progression of the stage of chronic hepatitis, and a study of 294 HBV patients in China reported that the HBcrAg level was 9.30 log U/mL in IT, 8.80 log U/mL in IC, 5.10 log U/mL in ENH, and 3.00 log U/mL in ENQ, and that the cut-off values between IC and IT and between ENQ and ENH were 9.25 log U/mL and 4.15 log U/mL, respectively [82].

In HBe antigen-negative CHB in particular, the HBcrAg level is known to correlate well with markers such as ALT, APRL, and FIB4 [83]. Seto et al. examined 404 untreated patients by classifying them into IT, IC, ENH, ENQ, and HBsAg seroclearance, and reported that the HBcrAg level correlated well with the HBVDNA level, particularly in the ENQ group, that the positive rate was also high at 36.4% in the HBsAg-negative group, and that this sensitivity was comparable to that of HBsAg-HQ (22.7%) [84]. Also, in a European cohort mainly from Germany, HBcrAg has been shown to reflect the disease status, with most cases in ENQ showing an HBcrAg level of less than 3.0 log U/mL [85]. Similarly, Loggi et al. also reported that in HBe antigen-negative chronic hepatitis B, HBcrAg is useful in differentiation of the gray zone between active CHB and clinically quiescent infection (CIB), suggesting 2.5 log U/mL as the optimal cut-off level [86].

### 3.3. HBcrAg and Antiviral Therapy

In a 96-week observational study in Hong Kong of 120 untreated patients treated with ETV and TDF, there was no significant difference in the reduction of HBcrAg (TDF vs. ETV: 2.28 vs. 1.65 log U/mL, *p* > 0.05), but both drugs markedly reduced HBcrAg compared to the levels before the beginning of the treatment in the HBeAg-negative group (0.83 vs. 0.54 log U/mL, *p* > 0.05) [87]. More recently, the risk of relapse of hepatitis after discontinuation of nucleic acid analogues (clinical relapse) has been reported to be reduced to HR 0.41 when the HBcrAg level is less than 3.0 log U/mL [88].

The additional clinical significance of the HBcrAg assay in CHB is not yet fully understood [83]. Also, as the false-positive rate in CHB is 9.3% and the false-negative rate is 12–35%, caution is considered necessary in interpreting the results [89].

The development of a high-sensitivity assay of HBcrAg (iTACT-HBcrAg) increased detectability in an antiviral-treated patient. The study reported the detection of HBcrAg in 121 patients receiving nucleot(s)ide analogues and found that the highly sensitive method was able to detect HBcrAg in 95.7% of patients compared to 75.2% with the conventional method [80].

### 3.4. HBcrAg as a Predictor of Hepatocarcinogenesis

Since HBcrAg correlates well with cccDNA in the liver, many reports have been published on HBcrAg as a predictive marker for hepatocarcinogenesis. Kaneko et al. suggested that the HBcrAg level 1 year after the beginning of treatment with a nucleos(t)ide analogue preparation contributes to the risk assessment of carcinogenesis in HBe antigen-negative CHB patients and reported an HR of 6.749 with a cut-off level of 4.1 log U/mL [90]. Similarly, by analyzing 1400 CHB patients taking nucleotide analogues, Liang et al. reported that the risk of carcinogenesis increases 2.13-fold with an HBcrAg cut-off level of 2.9 log U/mL in HBe antigen-negative patients [91]. In an observational study of 1108 cases of HBV-associated cirrhosis and 219 cases of advanced HCC in Taiwan with a median follow-up period of 6.85 years, Chang et al. found that HR of the development of HCC was 1.70 with HBcrAg of 3.4–4.9 log U/mL and 2.14 with HBcrAg of >4.9 log U/mL compared with HBcrAg of ≤3.4 log U/mL, and that risk assessment using HBcrAg was useful, particularly when the HBsAg-HQ level was <3 × 10^7^ mIU/mL [92]. Tseng et al. reported that HBcrAg is an independent risk factor for the development of hepatocarcinoma and that patients with an intermittent viral load of HBVDNA 2000–19,999 IU/mL have an increased risk of carcinogenesis if HBcrAg is 4.0 log U/mL even with a normal ALT level [93]. It has also been reported that HBcrAg can be a risk for hepatocarcinogenesis even if HBVDNA is below the detection sensitivity limit during antiviral therapy [94,95,96,97]. Hosaka et al. observed 1268 chronic hepatitis B patients treated with nucleic acid analogues for more than 1 year for a median follow-up period of 8.9 years and reported that 113 cases developed HCC, that the HBcrAg level during treatment is a risk factor for the progression of HCC, and that HR was 6.15 with a cut-off level of 4.9 log U/L in HBe antigen-positive patients and 2.54 with cut-off level of 4.4 log U/mL in HBe antigen-negative patients [96]. In Taiwan, 11.1% of patients with nonB–nonC HCC were HBcr antigen-positive, indicating that examination of HBcrAg is effective for risk screening in HBV endemic areas [94]. Recently, Hosaka et al. reported using a high-sensitivity HBcrAg assay system that the lower the measured value of HBcrAg 1 year after the beginning of oral administration of nucleos(t)ide analogues, the lower the incidence of HCC [98].

### 3.5. HBV Reactivation and HBcrAg

Recently, a high-sensitivity HBcr antigen assay system (iTACT-HBcrAg, cut-off value: 2.1 log IU/mL) has become available [80]. Its use reduced the lower limit of assay, which used to be 2.9 log U/mL, to 2.0 (Table 2). Hagiwara et al. evaluated the usefulness of the assay system using actual samples from patients with HBV reactivation (defined as HBVDNA levels of 1.3 log IU/mL or more) and who received systemic chemotherapies for hematologic malignancies. The results showed that iTACT-HBcrAg had a significantly higher detection rate than HBsAg-HQ (96% vs. 52%, respectively) and suggest that the HBcrAg level may become a surrogate marker for HBV reactivation [99]. Recently, the performance of iTACT-HBcrAg has been compared with that of a high-sensitivity HBs antigen assay system (iTACT-HBsAg) using samples from patients with HBV reactivation, indicating a higher sensitivity of iTACT-HBcrAg than iTACT-HBsAg (96.3% vs. 88.9%) [100].

## 4. Remaining Issues for the Future

By improving the measurement system, anti-HBc antibodies can be measured with high sensitivity. However, the risk of false positives increases with ultrahigh sensitivity, so the challenge is how to reduce the number of false positives. Once diagnosed as anti-HBc antibody-positive, risk management concerns and regular testing is required for patients undergoing treatment with a risk of reactivation. As for HBcrAg, a highly sensitive measurement system has been developed, but clinical evaluation is immature due to its short period of use. It is expected that the measurement system of HBcrAg will be more widely available and the clinical significance of HBcrAg will be strenthened in the future, including the evaluation of the efficacy of new antiviral therapies (Table 3). Capsid assembly modulators (CAMs) targeting the core region have been shown to suppress pregenomic RNA and reduce cccDNA [101,102]. CAMs are expected to be novel therapeutic agents against HBV [103,104,105,106,107]. It is hoped that knowledge of the HBV core region will continue to increase in the future.

## 5. Conclusions

With the development of highly sensitive HBcrAg and anti-HBc antibody, a rapid and sensitive detection assay system has been developed and is used in clinical practice. In the future, it is hoped that a global standard will be created based on the many clinical findings.

## Figures and Tables

**Table 1 diagnostics-14-00728-t001:** Common anti-HBc antibody assay techniques.

Product Name	Manufacturer	Method	Positive Range	Unit	Refs.
ADVIA Centaur^®^ HBc Total 2 assay	Bayer Diagnostics (Tarrytown, NY, USA)	CLIA	≥1.0	Index	[9,10]
MUREX anti-HBc	DiaSorin (Saluggia, Italy)	ELISA			[11]
MONOLISA anti-HBc EIA	Bio-Rad (Hercules, CA, USA)	EIA	≥0.5	Index	[11,12]
ORTHO HBc ELISA test	Ortho-Clinical Diagnostics (Raritan, NJ, USA)	ELISA			[13]
Lumipulse HBcAb-N^®^	Fujirebio (Tokyo, Japan)	CLEIA	≥1.0	COI	[14]
Elecsys^®^, Anti-HBc	Roche Diagnostics (Basel, Switzerland)	ECLIA	<1.0	COI	[15,16]
CLIA Microparticles	Autobio Diagnostics, Zhengzhou, China	CLIA			[17]
ARCHITECT^®^ HBc II	Abbott Diagnostics (Chicago, IL, USA)	CMIA	≥1.0	S/CO	[18]
HBcAb ELISA	Wantai Biological Pharmacy Enterprise Company (Beijing, China)	ELISA	≥1.0	IU/mL	[19]
	InnoDx Biotech (Xiamen, China)	CMIA		IU/mL	[20]

CLIA: chemiluminescent immunoassay; ELISA: enzyme-linked immunosorbent assay; EIA: enzyme immunoassay; CLEIA: chemiluminescent enzyme immunoassay; ECLIA: electrochemiluminescence immunoassay; COI: cut-off index; S/CO: sample cut-off; IU: international units.

**Table 2 diagnostics-14-00728-t002:** Comparison of Lumpulse HBcrAg and Lumpulse presto iTACT HBcrAg.

	Lumpulse HBcrAg	Lumpulse Presto iTACT HBcrAg	Ref.
Method	Two-step sandwich assay (CLEIA)	Two-step sandwich assay (CLEIA)	
Sample volume	150 μL	50 μL	
Measurement time	65 min	35 min	
Range of measurement	3.0 to 7.0 logU/mL	2.1 to 7.0 logU/mL	
Target machine	Lumpulse G600II/Lumpulse G1200	Lumpulese L2400	
Detectability in antiviral-treated patients	75.2%	97.5%	[80]

CLEIA: chemiluminescent enzyme immunoassay.

**Table 3 diagnostics-14-00728-t003:** Clinical utility of the anti-HBc antibody and the HBcrAg in the different clinical situations, the recommended method, and their significance of positivity.

	Anti-HBc Antibody	HBcr Antigen
Timing of measurement	Screening during blood donation Screening before immunosuppressive therapy	Estimation of cccDNA content in cases of HBs antigen-positive Purpose of evaluating treatment efficacy during HBV antiviral treatment
Methods and purposes	Various, such as ELISA, EIA and CLEIA Highly sensitive assay such as CLEIA is recommended for close examination, but simple tests such as EIA are useful for epidemiological studies	Standardized method by CLEIA
Clinical significance of positive result	Past or present infection In quantitative methods, the present infection is generally high-titer	Evaluation of antigen levels of precure mRNA and p22cr antigen and HBc antigen in combination
Significance of monitoring	Risk assessment of reactivation in immunosuppressed patients	Prediction of treatment efficacy and carcinogenesis during antiviral therapy Risk assessment of reactivation in immunosuppressed patients
Clinical significance of negative result	No history of infection False negative is possible in immunosuppressed status	No history of infection Antigen production is decreased and hepatitis is quiescent
Future issues	Uniformity of units	Widespread use of measuring instruments Clinical evaluation is immature due to short period of use

ELISA: enzyme-linked immunosorbent assay; EIA: enzyme immunoassay; CLEIA: chemiluminescent enzyme immunoassay; cccDNA: covalently circular closed DNA.

## Data Availability

Data are contained within the manuscript.

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
