# Peer review of "Clinical Significance and Remaining Issues of Anti-HBc Antibody and HBV Core-Related Antigen"

_diagnostics, 2024, doi:10.3390/diagnostics14070728_

Round 1

Reviewer 1 Report

Comments and Suggestions for Authors

The manuscript: "Clinical significance and remaining issues of anti-HBc antibody and HBV core-related antigen" is a well written review of the current methods for quantification and clinical relevance of anti-HBc and HBcrAg. There are just a couple of changes the authors should make in order for manuscript to be suitable for publishing. 1. I believe the reference number 19 is missing from the text, even though it has been listed under References. Line 67 lists references 8-18, then you list individual references, the last one is 15 in line 88, and then you have reference number 20 in line 90. If I have missed the reference 19 somewhere I apologize. 2. There should not be any references in the Conclusion part, it should only state what you have already confirmed by references in the text before. I believe the paper will be suitable for publishing after correcting the said errors.

Comments on the Quality of English Language

The English language requires only minor editing.

Author Response

Thank you for your careful suggestion. I apologize our mistake and agree with correction. As reviewer pointed out, the reference in the text should be changed to [9-20] from [8-18].

Reviewer 2 Report

Comments and Suggestions for Authors

The study by Yano Y et al. is a narrative review focused on hepatitis B virus (HBV) core antibody (anti-HBc antibody) and HBV core-related antigen (HBcrAg). The authors review the methods and units, general clinical significance, and the usefulness of anti-HBc and HBcrAg in different clinical situations. 

This narrative review is interesting. However, significant changes are necessary to improve the qualitative and for a better understanding.

Major comments

1.     Abstract. Please include the aim, the methodology, and the conclusions of this review.

2.     Introduction. Please include the aim at the end of the introduction section.

3.     Methods. Please include a new paragraph explaining the methodology used in this narrative review.

4.     Line 53. Please include the meaning of the S/CO ratio and log IU/mL.

5.     Sections. Please organize the title and the content with a more comprehensive view of the virology of anti-HBc and HBcrAg, the different methods and units, and the clinical significance in different situations.

6.     Section 2.2 and Table 1. Please include the range of detection, sensitivity, and specificity of each method

7.     Section 2.4. Please pull apart the information about acute and chronic hepatitis B and the information regarding the natural course and the antiviral treatment.

8.     Section 2.6. Please include this section as part of the chronic hepatitis.

9.     Section 2.7. Please include information about other transplantations.

10.  Table 2. Please include the sensitivity and specificity of each method.

11.  Section 3.2. Please include information regarding HBeAg-positive patients. Please pull apart the information regarding the natural course of hepatitis B and the antiviral treatment.

12.  Section 3.4. Please explain in more detail the clinical situation of “reactivation”.

13.  Other recommendations. Please include one or two new table/s summarizing the utility of the anti-HBc and the HBcrAg in the different clinical situations, the recommended method, and their cut-offs.

14.  Discussion. Please summarize in a new paragraph the anti-HBc and HBcrAg utility in clinical practice, the special considerations, and the future directions.

15.  Conclusion. Please change this conclusion to a new one related to the aim and content of the review.

Author Response

Thank you for precise suggestions. I agree with all suggestions and correct the manuscript. add the aim and conclusion in the abstract.

Aim in this study: The diagnostic significance of blood tests is changing with the development of many HBV assay systems and changes in disease management. This review focused on anti-HBc antibodies and HBcrAg and aimed to summarize the clinical significance of currently used assay systems and the issues involved.

Conclusion in this study: With the development of highly sensitive HBcrAg and HBcAb, a rapid and sensitive detection assay system has been developed and used in clinical practice. In the future, it is hoped that a global standard will be created based on the many clinical findings.

Especially, we add table 3 about summarizing the utility of the anti-HBc and the HBcrAg in the different clinical situations et. In addition, we add the paragraph about “4. Remaining issues for the future” section.

As for sensitivity and specificity of each measurement system, it is difficult to describe because they vary greatly depending on the pathological condition, so we have added them to the extent possible.

Round 2

Reviewer 2 Report

Comments and Suggestions for Authors

The authors have made modifications to the original manuscript based on the reviewers’ comments and advice improving the quality of their study.

Now, the manuscript is suitable for publication in diagnostics